# Genetic *LGALS1* Variants Are Associated with Heterogeneity in Galectin-1 Serum Levels in Patients with Early Arthritis

**DOI:** 10.3390/ijms23137181

**Published:** 2022-06-28

**Authors:** Ana Triguero-Martínez, Emilia Roy-Vallejo, Nuria Montes, Hortensia de la Fuente, Ana María Ortiz, Santos Castañeda, Isidoro González-Álvaro, Amalia Lamana

**Affiliations:** 1Rheumatology Department, Hospital Universitario La Princesa, Instituto de Investigación Sanitaria La Princesa (IIS-IP), 28006 Madrid, Spain; ana6n92@gmail.com (A.T.-M.); nuria.montes.casado@gmail.com (N.M.); lanult@yahoo.es (A.M.O.); scastas@gmail.com (S.C.); 2Internal Medicine Department, Hospital Universitario La Princesa, Instituto de Investigación Sanitaria La Princesa (IIS-IP), 28006 Madrid, Spain; eroyvallejo@gmail.com; 3Immunology Department, Hospital Universitario La Princesa, Instituto de Investigación Sanitaria La Princesa (IIS-IP), 28006 Madrid, Spain; hortensiadelafuente@gmail.com; 4Cell Biology Department, Facultad de Biología, Universidad Complutense de Madrid, 28040 Madrid, Spain

**Keywords:** rheumatoid arthritis, galectin-1, *LGALS1* gene, biomarker, early arthritis, single nucleotide polymorphism

## Abstract

Galectin 1 (Gal1) exerts immunomodulatory effects leading to therapeutic effects in autoimmune animal models. Patients with rheumatoid arthritis have been reported to show higher Gal1 serum levels than the healthy population. Our study aimed to find genetic variants on the Gal1 gene (*LGALS1*) modulating its expression and/or clinical features in patients with early arthritis (EA). *LGALS1* was sequenced in 53 EA patients to characterize all genetic variants. Then, we genotyped rs9622682, rs929039, and rs4820293, which covered the main genetic variation in *LGALS1*, in 532 EA patients. Gal1 and IL-6 serum levels were measured by ELISA and Gal1 also by western blot (WB) in lymphocytes from patients with specific genotypes. Once disease activity improved with treatment, patients with at least one copy of the minor allele in rs9622682 and rs929039 or those with GG genotype in rs4820293 showed significantly higher Gal1 serum levels (*p* < 0.05). These genotypic combinations were also associated with higher Gal1 expression in lymphocytes by WB and lower IL-6 serum levels in EA patients. In summary, our study suggests that genetic variants studied in *LGALS1* can explain heterogeneity in Gal1 serum levels showing that patients with higher Gal1 levels have lower serum IL-6 levels.

## 1. Introduction

Rheumatoid arthritis (RA) is the most common autoimmune disease, causing polyarthritis with a prevalence of about 1% in the general population [1]. RA is characterized by infiltration of immune cells to the synovial membrane triggering irreversible joint damage, although it can also cause systemic manifestations. RA pathogenesis is complex, involving interactions between multiple genetic and no genetic risk factors [2]. Genome-wide association studies (GWAS) have been the principal tool to study complex diseases like RA and, at present, it has been identified around 101 RA susceptibility loci related to the risk of developing RA [3]. However, the individual contribution of each locus is low, and there is little information about the relevance of the interaction between genetic factors, and even less is known about the role of genetic variants in the severity of RA. In this sense, numerous studies have focused on looking for genetic variants involved in molecules that play an important role in RA physiopathology or factors related to disease severity.

Galectins are lectins that bind β-galactosides carbohydrates [4] and they are among the best-studied glycan-binding proteins implicated in the regulation of inflammatory responses [5]. Galectins can participate in immune system homeostasis and rearrange several signaling pathways in different immune cells and tissues, thus modulating the inflammatory response [6]. Galectin-1 (Gal1) has been described to be involved in autoimmunity and chronic inflammation, mainly with regulatory functions [5]. The first data about the role of Gal1 in RA was provided by Rabinovich et al. showing its therapeutic effect in a murine model of collagen-induced arthritis (CIA) [7]. The same group observed increased Gal1 serum levels in patients with prevalent RA [8]. Recently, we confirmed that patients with RA display increased Gal1 serum levels and we proposed Gal1 as a possible biomarker in RA [9]. However, significant heterogeneity in the levels of Gal1 serum was observed in RA patients that could not be explained by variations in their level of disease activity [9].

*LGALS1* gene is located on chromosome 22q12 [10]. Although one study has demonstrated that rs4820294 and rs13057866 in *LGALS1* are associated with higher expression of Gal1 protein in human cells and decreased susceptibility to influenza infection [11], there are few/no studies showing an association between *LGALS1* genetic variants and Gal1 serum levels. Therefore, the objective of this work was to study the effect of genetic variants in *LGALS1* on Gal1 expression and its relationship with clinical features in patients with early arthritis (EA).

## 2. Results

### 2.1. Study of Genetic Variability in the Gal1 Promoter and Gene

The first objective of this work was to sequence the promoter and the complete *LGALS1* gene (introns and exons) in 53 EA patients to identify tag single nucleotide polymorphisms (SNPs) that would allow us to infer the genetic variability of *LGALS1* and that had minor allele frequencies (MAF) > 0.1 in order to test them as biomarkers in arthritis (Appendix A). This approach allowed us to detect 20 SNPs already described, of which only 11 showed a MAF > 0.1 in EA patients (Appendix A).

After performing a linkage disequilibrium (LD) analysis (Appendix A), we chose rs9622682 (located in intron 6) as the first candidate for showing high LD with the main SNPs of the *Gal1* gene (D′ ≤ 1.0) and being quite polymorphic (MAF = 0.488). Also, we selected rs929039 and rs4820293, which are located in the promoter region (upstream transcript variant) and showed MAF C = 0.30 and A = 0.38 respectively. These SNPs were selected as tag SNPs in our study population and for genotyping in our entire early arthritis population.

### 2.2. Effect of rs929039, rs9622682, and rs4820293 in Gal1 Serum Levels along the Follow-Up

To determine the role of genetic variability in *LGALS1* upon Gal1 serum levels, we analyzed the Gal1 serum levels data from 198 visits corresponding to 60 EA patients considering their genotype for rs929039, rs9622682, and rs4820293 and the visits in the follow-up.

As it is shown in Figure 1, in visits 2nd to 4th, when disease activity had improved with treatment (Appendix A), those patients with at least one minor allele from rs929039 or rs9622682 showed a significantly higher Gal1 serum levels (*p* = 0.05 and *p* = 0.02 respectively, Figure 1). By contrast, those patients who carried at least one minor allele rs4820293 displayed significantly lower Gal1 serum levels (*p* = 0.03, Figure 1). Interestingly, at baseline, when disease activity was higher (Appendix A), there were no differences in Gal1 serum levels according to the genotypes rs929039, rs9622682, or rs4820293.

Considering the opposite effects of the minor alleles of rs929039, rs9622682 vs rs4820293, we studied the effect of the combination of these genotypes on Gal1 serum levels in EA patients (Figure 1D). As expected, those patients with, at least one minor allele of rs929039 and rs9622682 and homozygous for the major allele of rs4820293 (genotype combination 1) exhibited significantly higher levels of Gal1 serum than those with, at least one minor allele of rs4820293 and homozygous for the major allele of rs929039 and rs9622682 (genotype combination 2, *p* = 0.0019).

Finally, there were no significant differences in disease activity assessed by the DAS28 score according to the genotype of the SNPs studied, neither when patients were active at baseline nor when disease activity was controlled after treatment prescription (Appendix A). Therefore, we can conclude that genotypes from rs929039, rs9622682, and rs4820293 SNPs could be influencing Gal1 serum levels in EA patients but have no clear relationship with disease activity.

### 2.3. rs929039, rs9622682, and rs4820293 Genotypes Are Associated with Gal1 Expression in Peripheral Blood Lymphocytes

To further characterise the role rs929039, rs9622682, and rs4820293 in Gal1 protein expression, we evaluated its levels by western blot (WB) in lymphocytes isolated from peripheral blood of 86 EA patients selected according to the presence of homozygous genotypes from the SNPs rs929039 (TT/CC), rs9622682 (GG/AA) and rs4820293 (GG/AA).

Lymphocytes from EA patients carrying the CC (rs929039), AA (rs9622682), or GG (rs4820293) genotype showed significantly higher Gal1 expression compared with EA patients carrying the GG, TT, or AA genotypes respectively (Figure 2, panels A–C).

When we analysed these genotypes combined (as explained for Gal1 serum levels), patients with genotype combination 1 had higher Gal1 than the patients from genotype combination 2 and this result validated what we observed on an individual analysis (Figure 1D).

### 2.4. The Genotypes rs929039, rs9622682, and rs4820293 Are Associated with IL-6 Serum Levels in EA Patients

It has been described that Gal1 can block the in vitro secretion of pro-inflammatory cytokines (IL-2, IFN and TNFα) [12] and specifically that Gal1 decreases IL-6 production [13], the latter playing an important role in RA. Moreover, our group showed in a previous study that Gal1 correlated with IL-6 in the serum of RA patients [9]. Therefore, we decided to explore the effect of *LGALS1* genetic variants on IL-6 serum levels.

For this purpose, we performed a multivariate analysis in which we adjusted IL-6 serum levels by disease activity, since it is well known to correlate with it, and by methotrexate treatment, which we have previously described to independently decrease IL-6 levels [14,15]. Our multivariate analysis revealed that carrying at least one minor allele for SNPs rs929039 or rs9622682 was significantly associated with lower IL-6 serum levels (Table 1 and Figure 3A,B). On the other hand, the minor allele of rs4820293 was associated with the opposite effect, as patients carrying minor alleles showed higher serum IL-6 regardless of disease activity levels (Figure 3C and Table 2). In addition, we also analysed the effect that the combination of genotypes could have on IL-6 serum levels (genotype combinations 1 and 2, above described, Figure 3D), and we observed that patients with genotype combination 1 had lower IL-6 serum levels than genotype combination 2. 

On the other hand, genetic variants in the promoter region of the *IL-6* gene (rs1800795) and in IL-6 receptor (IL-6R) (rs2228145) may exert some effect on the heterogeneity of serum IL-6 levels [16,17,18,19]. Therefore, we also analyzed the effect of these SNPs in the previous multivariate analysis. The presence of at least one C allele of rs2228145 was associated with a significant increase in serum IL-6 levels (Appendix A). In contrast, rs1800795 genotypes were not significantly associated with serum IL-6 levels (Appendix A).

Despite adjusting for the rs2228145 and rs1800795 genotypes, the effect of Gal1 SNPs remained highly significant. Therefore, the association of the 3 SNPs described in *LGALS1* is not due to collinearity, it is an independent effect.

Furthermore, all multivariable analyses were adjusted for the level of disease activity. 

## 3. Discussion

Rheumatoid arthritis has a complex physiopathology. The interaction of genetic, environmental, psychosocial and stochastic factors causes the rupture of self-tolerance leading to a systemic autoimmune disorder in which diarthrodial joints are the main target, although other organs can be involved [1]. Currently, the stages that cause the disease are not fully elucidated, and probably are different from one patient to another. Regarding the genetic basis of the disease, there are more than 100 loci associated with the risk of developing RA, some of which are shared with other chronic inflammatory diseases [3]. However, there is not enough information about the role of genetic factors underlying the huge heterogeneity in the clinical burden and severity of RA. In this regard, the main finding of our work is that those patients with the genotype combination including TT for rs929039, GG for rs9622682, and GA or AA for rs4820293 show significantly lower Gal1 serum levels and likely due to lower immunoregulatory effects, those patients display significantly higher IL-6 serum levels. Although, *LGALS1* has not been described as a risk factor locus for RA development, previous studies from our group have shown that RA severity can be modulated by genetic variants in loci associated with RA development, such as rs7574865 in STAT4 [15], or in other loci not related with RA development, such as rs688136 in VIP [20] or rs4780355 in SOCS1 [21]. Therefore, it makes sense that, due to their immunomodulatory effects, genetic variations in *LGALS1* involving Gal1 expression can regulate disease activity once the disease has started rather than be involved in the impaired self-tolerance leading to the development of RA.

It is commonly proposed that Gal1 is a soluble protein that functions in the extracellular environment by interacting with glycosylated receptors or intracellularly by controlling signaling pathways through protein-glycan or protein-protein interactions [22]. However, the role of Gal1 on IL-6 production is controversial with studies suggesting Gal1 as an inducer [23,24,25] whereas others suggest that Gal1 decreases IL-6 production [13,26,27,28]. Our data support this last proposition since even adjusting for disease activity, genetic variants in IL-6 or IL-6R or methotrexate dose, *LGALS1* variants associated with higher Gal1 serum levels were associated with lower IL-6 levels.

It is important to keep in mind that many extracellular functions of galectins depend on the action of glycosyltransferases and glycosidases, which regulate the availability of N- and O-glycan patterns and are controlled by cytokines, hypoxia, and inflammation, therefore the effect of galectins can vary depending on the experimental model and in vitro conditions [29,30]. In this regard, O-GlcNAcylation has been recently unveiled as a new mechanism controlling galectin secretion [31,32,33].

Regarding RA, our work supports an anti-inflammatory effect of serum Gal1, since we observed that patients with at least one minor allele of rs929039 and rs9622682 and homozygous for the major allele of rs4820293 have higher Gal1 and lower IL-6 serum levels. Interestingly, we were unable to detect significant differences in Gal1 levels when patients showed a high disease activity at the beginning of the follow-up. This finding suggests that genetic variations in Gal1 induce subtle regulatory effects that cannot be detected when intense activation by pro-inflammatory stimuli occurs. However, the effect of SNPs became apparent when disease activity was controlled by treatment. In this regard, we were also unable to detect differences in lymphocytes Gal1 expression when they were treated with different stimuli (data not shown), whereas differences according to rs929039, rs9622682, and rs4820293 specific genotypes were detected in resting lymphocytes.

Another intriguing aspect of our data is that we did not detect differences in disease activity between patients according to the different genotypes of the SNPs studied. A possible explanation is that PEARL is an observational study without a pre-established therapeutic schedule. In addition, the “treat to target” strategy is widely assumed by rheumatologists in our Unit [34]. Therefore, physicians would have intensified the treatment until the patient achieves remission or low disease activity. This can have masked differences in the severity of RA between patients with different genotypes.

On the other hand, considering that after treatment prescription those patients with low Gal1 serum levels displayed high IL-6 serum levels, those with inadequate control of the disease with standard disease-modifying antirheumatic drugs might be considered candidates to intensify treatment through IL-6 signaling blockade, an effective tool in the treatment of RA patients [35]. In this context, Gal1 levels could be useful as a possible biomarker to classify those patients that could benefit from IL-6 blocking therapy.

Finally, our study has some limitations. First, the SNPs rs929039, rs9622682, and rs4820293 were selected as tag SNPs capturing the main variability in *LGALS1*. Therefore, it is not clear whether they cause a direct effect on the regulation of *LGALS1* transcription or whether this effect is caused by other SNPs in linkage disequilibrium with those chosen in our study. In this regard, it is interesting that rs929039 and rs4820293 are located in the promoter region of *LGALS1*. Additional studies would be necessary to know how genetic variation in the *Gal1* gene results in differences in transcription, mRNA survival, etc., leading to differences in Gal1 production or secretion. Second, this is a one-centre study in which rheumatologists tend to be aggressive in the control of RA. Therefore, validation studies in other cohorts are needed.

## 4. Materials and Methods

### 4.1. Study Population

The present study was performed with data and samples from the Princesa Early Arthritis Register Longitudinal (PEARL) study carried out at the Hospital Universitario La Princesa, Madrid, Spain. The register protocol includes 4 visits during a 2-year follow-up (0, 6, 12, and 24 months). Socio-demographic, clinical, therapeutic and laboratory data are recorded and included in an electronic database. Biological samples (serum, DNA, and mRNA) are collected at each visit and stored at −80 °C in the Instituto de Investigación Sanitaria La Princesa (IIS-IP) Biobank (Madrid, Spain) for translational research. A more detailed description of the PEARL protocol has been previously published [36].

At the end of follow-up, all EA patients are classified as RA if they fulfil the 1987 ACR classification criteria [37] or as undifferentiated arthritis (UA) as described by Verpoort et al. [38]. Patients with other defined diagnoses (connective tissue diseases, psoriatic arthritis, gout, or osteoarthritis) are excluded from the study.

For this study, 532 patients from the PEARL study were considered although there are variations in the number of patients included in each experimental approach, so it will be specified in each section of methods, results, and figures.

Patients were clustered into two populations; in population 1 we selected 53 patients to sequence the *LGALS1* gene as a representation of the whole population. Population 2 comprised 532 patients EA patients who were genotyping to rs929039, rs9622682, and rs4820293. The characteristics of the two populations are shown in Table 1.

### 4.2. DNA Isolation and LGALS1 Sequencing

Genomic DNA was isolated from whole blood samples using the QIAamp DNA Blood Midi Kit (QIAGEN, Hilden, Germany).

For sequencing analysis, amplification primers were designed to produce overlapping polymerase chain reaction (PCR) amplicons to cover the region between positions 37675606 and 37679802 (GRCh38.p7) of chromosome 22, which includes the promoter and the gene encoding Gal1. A total of 26 PCR primers pairs were designed, including the M13 universal primer sequence (M13 forward primer sequence, 5′- TGTAAAACGACGGCCAGT-3′; and M13 reverse primer sequence 5′- CAGGAAACAGCTATGACC-3′) to generate 12 amplicons ranging between 463 bp and 770 bp (Appendix A).

PCR amplification and Sanger sequencing reactions were performed using the BigDye Direct Cycle Sequencing kit (Applied Biosystems, Waltham, MA, USA). Sequencing reactions were purified using the BigDyeXTerminator Purification kit and sequencing by 3500xLGenetic Analyzer (Applied Biosystems). The data were analyzed by comparing the results obtained with the consensus sequence of the *LGALS1* gene (NC_000022.11) using the analysis program Variant Reporter, version 1.1 (Applied Biosystems).

To quantify and compare linkage disequilibrium we use D′ values, which is the coefficient of LD between alleles located at different sites. D′ = 1 means that given the allele frequencies at the two sites, the allelic association is as strong as possible (complete LD) [39]. For the construction of the LD plot and the definition of the haplotypic block, we followed the model of Gabriel et al. [40]. All these calculations and graphs were performed with the Haploview program (Broad Institute, Cambridge, MA, USA).

### 4.3. rs9622682, rs929039, and rs4820293 Genotyping

A reliable genotype for rs9622682, rs929039 and rs4820293 was obtained in 523, 521 and 503 patients respectively using a pre-designed single nucleotide polymorphism (SNP) Genotyping Taqman Assays (Applied Biosystems, Part number: C___2495799_10, C___8957787_20, C___2495792_10 respectively). The PCR assay was carried out according to the manufacturer’s recommendations. After PCR, the genotype of each sample was determined automatically by measuring allele-specific fluorescence on a CFX Touch Real-Time PCR System using the software CFX 3.1 Manager (BioRad, Hercules, CA, USA). Duplicate samples and negative controls were included to verify genotyping accuracy.

### 4.4. Galectin-1 Assessment through Western-Blot

Peripheral blood samples from 47 early arthritis patients with specific homozygous genotypes for the SNPs of interest were used to analyse Gal1 expression in lymphocytes. First peripheral blood mononuclear cells were isolated by density gradient (Biocoll Separating Solution; Biochrom). Then, the monocytes were purified by positive selection using CD14 MicroBeads and autoMACS (both from Miltenyi Biotec) following the manufacturer’s instructions. T lymphocyte enriched fractions were lysed at 4 °C (during 30 min) in Tris-buffered saline (50 mM Tris-Cl, pH 7.5, 150 mM NaCl) 1% NP40 with a protease inhibitor cocktail (Roche Diagnostics, Indianapolis, IN, USA). Whole lysates were analysed by sodium dodecyl sulfate–polyacrylamide gel electrophoresis (SDS-PAGE), transferred to PVDF membranes and probed with the Human Galectin-1 antibody (R&D Systems) and the anti-GAPDH antibody FF26A/F9 (BioLegend, San Diego, CA, USA) in Tris buffered saline–Tween 20. Bound antibodies were conjugated with horseradish peroxidase secondary antibodies, and membranes were developed by enhanced chemiluminescence with Super-Signal West Femto chemiluminescent substrate (Pierce Chemical, Dallas, TX, USA). Densitometric analyses were performed with ImageGauge 3.46 software (Fujifilm, Greenwood, SC, USA).

### 4.5. Measure of Gal1 Serum and IL-6 in EA Population

Sixty patients with at least three visits along the two years follow-up (198 visits; 3.3 visits per patient) were selected to measure Gal1 serum levels in EA patients using Quantikine Human Gal1 Immunoassay (R&D Systems, Minneapolis, MN, USA).

IL-6 serum levels were measured using Human IL-6 Quantikine high sensitivity enzyme-immune assay (R&D Systems) in samples from 225 EA patients with, at least, three visits along with the two years follow-up (861 visits; 3.8 visits per patient).

The procedures were carried out according to the manufacturer’s instructions. Absorbance was measured in a spectrophotometer (Innogenetics Diagnostica y terapeutica S.A.U, Barcelona, Spain) at 450 nm with correction at 620 nm. Measurements for all samples were performed in duplicate.

### 4.6. Statistical Analysis

Statistical analyses were performed using Stata 14.0 for Windows (Stata Corp LP, College Station, TX, USA). Most quantitative variables followed a non-normal distribution, so they were represented as the median and interquartile range (IQR), and the Mann Whitney or Kruskal-Wallis tests were used to analyse significant differences. Qualitative variables were described as proportions, and the χ^2^ or Fisher’s exact test was used to compare categorical variables.

As described previously Gal1 serum levels significantly increased with age [41]. Therefore, we have considered this data when carrying out the statistical analysis.

### 4.7. Multivariate Analyses

In order to evaluate if the genetic variants from *LGALS1* (rs9622682, rs929039, and rs4820293) could be associated with variation in Gal1 or IL-6 serum levels we fitted a multivariate analysis using generalised linear models nested by patient and visits using the *xtgee* command of STATA. The population-averaged generalised estimating equations were first modelled by adding all variables with a *p* value < 0.15 to the bivariate analysis. The final models were constructed using quasi-likelihood estimation based on the independence model information criterion [42] and Wald tests, removing all variables with *p* > 0.15. Once the best model was obtained, several variables related to the assessment of RA were forced into the model in order to determine whether there was an association with Gal1 and/or IL-6 or not. Those variables that do not follow a normal distribution have been mathematically transformed to achieve normality. IL-6 serum levels were transformed through the square root. The best model was obtained as described above.

## 5. Conclusions

Even though Gal1 serum levels are higher in RA patients compared to a healthy population, our study suggests that patients who carry at least one copy of the minor allele in rs9622682 and rs929039 genetics variants or those with GG genotype in rs4820293 tend to have higher Gal1 levels and lower IL-6 serum levels.

## Figures and Tables

**Figure 1 ijms-23-07181-f001:**
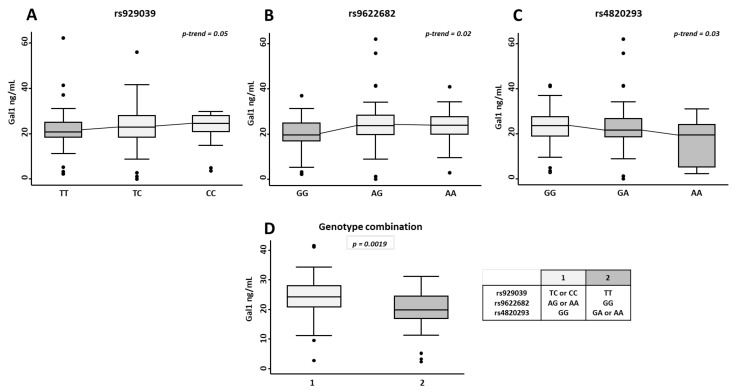
Effect of the *LGALS1* genetic variants rs929039, rs9622682, and rs4820293 in galectin 1 (Gal1) serum levels. Determination of Gal1 serum levels by ELISA in early arthritis patients (undifferentiated arthritis and rheumatoid arthritis) from the PEARL study according to the different genotypes for the single nucleotide polymorphisms (SNPs) rs929039 (**A**), rs9622682 (**B**), rs4820293 (**C**) and genotypes combination (**D**). Data are shown as interquartile range (p75 upper edge of box, p25 lower edge, p50 midline) as well as the p95 (line above box) and p5 (line below). Dots represent outliers. Statistical significance for the trend of Gal1 across the different genotype in patients was determined with Cuzick’s non-parametric test. The significance threshold was set at *p*-trend < 0.05.

**Figure 2 ijms-23-07181-f002:**
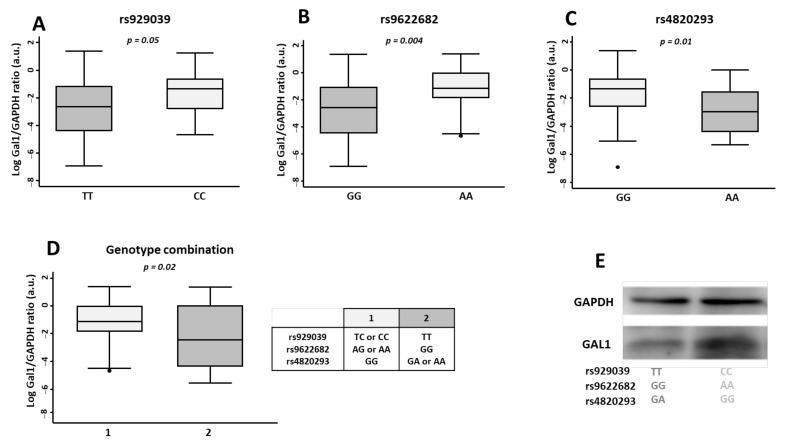
Effect of homozygous genotypes from *LGALS1* SNPs rs929039, rs9622682, and rs4820293 in protein expression from lymphocytes assessed by western blot. Densitometric quantification of galectin 1 (Gal1) protein expression normalised to GAPDH expression in 47 early arthritis patients from the PEARL study according to homozygous genotype from de genetics variants rs929039 (**A**), rs9622682 (**B**), rs4820293 (**C**), and genotypes combination (**D**). Data are shown as interquartile range (p75 upper edge of the box, p25 lower edge, p50 midline) as well as the p95 (line above box) and p5 (line below). Dots represent outliers. Statistical significance was determined with the t-student test. The significance threshold was set at *p* < 0.05. (**E**) Representative blot from Gal1 expression in EA patient’s lymphocytes.

**Figure 3 ijms-23-07181-f003:**
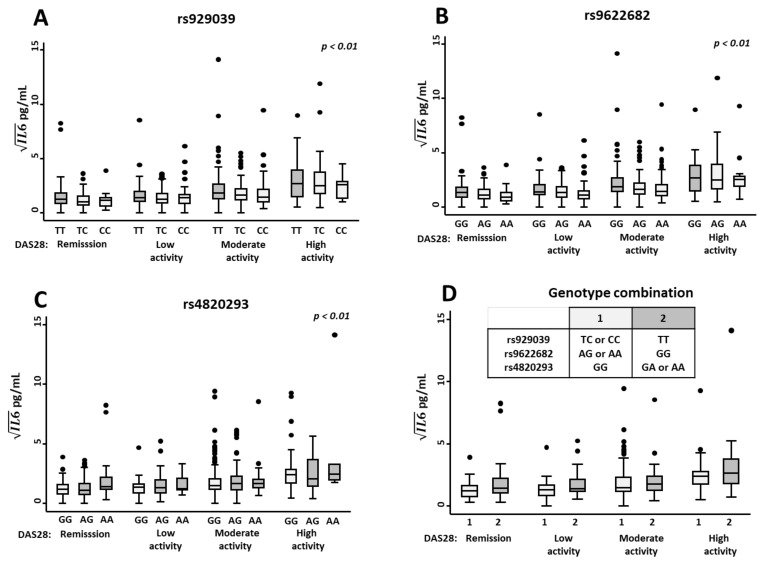
Association between genotypes of rs929039, rs9622682, and rs4820293 and IL-6 serum levels in early arthritis (EA) patients. Relationship between IL-6 serum levels (pg/mL) measured by ELISA in EA patients from PEARL study categorise by disease activity assessed by DAS28 score and the different genotypes of single nucleotide polymorphisms (SNPs) rs929039 (**A**), rs9622682 (**B**), rs4820293 (**C**) and genotype combination (**D**). Data are shown as interquartile range (p75 upper edge of box, p25 lower edge, p50 midline) as well as the p95 (line above box) and p5 (line below). Dots represent outliers. The statistical significance was determined with the multivariable analysis displayed in Table 2. The significance threshold was set at *p* < 0.05.

**Table 1 ijms-23-07181-t001:** Relationship between IL-6 (pg/mL) serum levels and *LGALS1* genetic variants.

	*LGALS1* (rs929039)		*LGALS1* (rs9622682)		*LGALS1* (rs4820293)	
	β Coeff. (95% CI)	*p* Value	β Coeff. (95% CI)	*p* Value	β Coeff. (95% CI)	*p* Value
**DAS28**						
Remission	Reference	−	Reference	−	Reference	−
Low DA	0.19 (0.05 to 0.33)	0.006	0.17 (0.03 to 0.31)	0.01	0.21 (0.06 to 0.36)	0.04
Moderate DA	0.61 (0.45 to 0.76)	<0.001	0.62 (0.46 to 0.78)	<0.001	0.66 (0.50 to 0.82)	<0.001
High DA	1.38 (1 to 1.76)	<0.001	1.39 (1 to 1.78)	<0.001	1.30 (0.93 to 1.68)	<0.001
**Methotrexate dose (mg)**	−0.008 (−0.01 to −0.0007)	0.03	−0.008 (−0.015 to −0.0002)	0.04	−0.01 (−0.018 to −0.002)	0.01
***LGALS1* (rs929039)**						
TT	Reference	−				
TC	−0.35 (−0.56 to −0.13)	0.001				
CC	−0.39 (−0.67 to −0.1)	0.007				
***LGALS1* (rs9622682)**						
GG			Reference	−		
GA			−0.38 (−0.63 to −0.13)	0.003		
AA			−0.52 (−0.78 to −0.26)	<0.001		
***LGALS1* (rs4820293)**						
GG					Reference	−
GA					0.14 (−0.03 to −0.33)	0.12
AA					0.58 (0.15 to 1)	0.007

DAS28: disease activity score estimated with 28 joint count; DA: disease activity; Coeff: coefficient; CI: confidence interval.

**Table 2 ijms-23-07181-t002:** Baseline clinical characteristics of the populations studied.

	Population 1	Population 2	*p*
	(*n* = 53)	(*n* = 479)	
Female; n (%)	38 (71.70)	384 (80.17)	0.15
Age; p50 [p25–p75]	53.62 [43.33–67.49]	55.16 [44.23–66.50]	0.69
Disease duration (months); p50 [p25–p75]	6.53 [4.06–8.53]	5.0.6 [2.76–8.5]	0.11
RF positive; n (%)	27 (50.94)	262 (54.70)	0.6
ACPA positive; n (%)	26 (49.06)	243 (51.16)	0.79
DAS28; p50 [p25–p75]	4.88 [3.85–6.05]	4.22 [3.21–5.51]	0.01
HAQ; p50 [p25–p75]	1.12 [0.62–1.62]	1 [0.43–1.62]	0.22

n: number; p50: median or percentile 50; p25–p75: range between percentiles 25 and 75 or interquartile range; RA: rheumatoid arthritis; UA: undifferentiated arthritis; RF: rheumatoid factor; ACPA: anti-citrullinated protein antibodies; DAS28: baseline disease activity score estimated with the 28 joint count; HAQ: baseline health assessment questionnaire.

## Data Availability

The datasets used and/or analysed during the current study are available from the corresponding author (isidoro.ga@ser.es and amaliala@ucm.es) on reasonable request.

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
