# Peer review of "Genetic LGALS1 Variants Are Associated with Heterogeneity in Galectin-1 Serum Levels in Patients with Early Arthritis"

_ijms, 2022, doi:10.3390/ijms23137181_

Round 1

Reviewer 1 Report

Paper by Triguero-Martínez et al. presents some evidences that three tag LGALS1 gene SNPs are associated to regulation of Gal1 serum level with an inverse relationship with IL-6 serum levels.

In addition, apparently, no association of the three tag SNP was found with disease activity.

There are some points that request attention.

Major:

1. Authors evaluated LGALS1 in a population of RA patients, identified tag SNP and then developed their experimental design. It is possible that this strategy presents some bias. Actually using a population of patients there is always the possibility that disease associated genetic background might affect  the relevance and frequency of some SNPs underscoring some potential “protective variants pumping up the “susceptibility” associated ones. In my opinion control subject DNA sequencing should be made to confirm or correct the results of LOD analyses.

2. Moreover it is possible that the analyses of the association of LGALS1 tag SNPs GAL1 levels and disease activity might have produced different results if an adequate number of control subjects were compared with patient with different DAS28 categorization

3. Authors indicated that the three SNP regulating GAL1 levels might influence IL-6 levels.

First of all, SNPs of IL-6 genes are known to modulate IL-6 level and production, so the interaction among LGALS1 tag SNPs and IL-6 functionally relevant SNP should be studied.

On the other hands, IL-6 is a good marker of active inflammation. So data obtained in RA patients might be interpreted as an association of GAL1 tag SNP and the presence of the disease activity.   Authors discuss this point but this should better explained.

Minor:

Some typo error should be corrected (e.g. taq SNPs instead of Tag SNP on page 2, last line of 2.1 result paragraph)

Reviewer 2 Report

The manuscript “ Genetic LGALS1 Variants are Associated with Heterogeneity in Galectin-1 Serum Levels in Patients with Early Arthritis” presents interesting findings that might be helpful to understand the manifestation of clinical features of patients with rheumatoid arthritis. The text is well-prepared and adequate methods were used to produce the results. Overall, the results are properly discussed and the authors mentioned a potential role of glycosyltransferases and glycosidases among other options in regulating the functions of galectins. This aspect can be articulated more specifically considering the role of O-GlcNAcylation as an emerging mechanism controlling secretion of galectins (PMID: 32731422). Although this mechanism has been not addressed yet for the secretion of galectin-1, strong experimental evidence has recently been reported for galectin-3 (PMID: 35625551; PMID: 35183508). There are also few minor typos to correct, e.g. abbreviation for rheumatoid arthritis (RA) is typed as AR in the introduction.

Round 2

Reviewer 1 Report

Authors have upgraded manuscript and satisfy almost all queries